# Broad Antiviral Effects of *Echinacea purpurea* against SARS-CoV-2 Variants of Concern and Potential Mechanism of Action

**DOI:** 10.3390/microorganisms10112145

**Published:** 2022-10-29

**Authors:** Selvarani Vimalanathan, Mahmoud Shehata, Kannan Sadasivam, Serena Delbue, Maria Dolci, Elena Pariani, Sarah D’Alessandro, Stephan Pleschka

**Affiliations:** 1Pathology & Laboratory Medicine, University of British Columbia, Vancouver, BC V6T 2B5, Canada; 2Institute of Medical Virology, Justus Liebig University Giessen, 35392 Giessen, Germany; 3Center of Scientific Excellence for Influenza Viruses, National Research Centre, Giza 12622, Egypt; 4Centre for High Computing, Central Leather Research Institute, Adyar, Chennai 600020, India; 5Laboratory of Molecular Virology, Department of Biomedical, Surgical and Dental Sciences, University of Milano, 20133 Milano, Italy; 6Department of Biomedical Sciences for Health, University of Milano, 20133 Milano, Italy; 7Department of Pharmacological and Biomedical Sciences, University of Milano, 20133 Milano, Italy; 8German Center for Infection Research, Partner Site Giessen-Marburg-Langen, 35392 Giessen, Germany

**Keywords:** Echinacea, SARS-CoV-2, variants of concern, antiviral, prevention, Spike protein, TMPRSS-2

## Abstract

SARS-CoV-2 variants of concern (VOCs) represent an alarming threat as they show altered biological behavior and may escape vaccination effectiveness. Broad-spectrum antivirals could play an important role to control infections. The activity of *Echinacea purpurea* (Echinaforce^®^ extract, EF) against (i) VOCs B1.1.7 (alpha), B.1.351.1 (beta), P.1 (gamma), B1.617.2 (delta), AV.1 (Scottish), B1.525 (eta), and B.1.1.529.BA1 (omicron); (ii) SARS-CoV-2 spike (S) protein-pseudotyped viral particles and reference strain OC43 as well as (iii) wild type SARS-CoV-2 (Hu-1) was analyzed. Molecular dynamics (MD) were applied to study the interaction of *Echinacea’s* phytochemical markers with known pharmacological viral and host cell targets. EF extract broadly inhibited the propagation of all investigated SARS-CoV-2 VOCs as well as the entry of SARS-CoV-2 pseudoparticles at EC_50′_s ranging from 3.62 to 12.03 µg/mL. The preventive addition of 25 µg/mL EF to epithelial cells significantly reduced sequential infection with SARS-CoV-2 (Hu-1) and OC43. MD analyses showed constant binding affinities to VOC-typical S protein variants for alkylamides, caftaric acid, and feruloyl-tartaric acid in EF extract and interactions with serine protease TMPRSS-2. EF extract demonstrated stable virucidal activity across seven tested VOCs, likely due to the constant affinity of the contained phytochemical substances to all spike variants. A possible interaction of EF with TMPRSS-2 partially would explain the cell protective benefits of the extract by the inhibition of membrane fusion and cell entry. EF may therefore offer a supportive addition to vaccination endeavors in the control of existing and future SARS-CoV-2 virus mutations.

## 1. Introduction

Severe acute respiratory syndrome corona virus (SARS-CoV-2) is an enveloped, ssRNA + sense genome, belonging to the order Nidovirale and the family of Coronaviridae. In general, the mutation rate is higher in RNA viruses compared to DNA viruses, due to the lack of proof-reading ability of polymerase enzyme with the exception of nidoviruses [1,2]. However, nidoviruses encode ExoN (Exorionuclease), which has the ability to correct errors occurring during replication. Despite the presence of ExoN in SARS-CoV-2, numerous genetically distinct lineages have evolved possibly due to the sheer number of infection/replication events. Furthermore, recent findings revealed events of inter-lineage recombination of different SARS-CoV-2 variants co-infecting a single cell [3,4]. Since the genomic decryption of WUHAN-Hu-1 in January 2020, the evolution of new genetic lineages has been tracked globally and so far, thousands of different genomes have been collected by the National SARS-CoV-2 Strain Surveillance (NS3) system [5]. Several new SARS-CoV-2 lineages are differing in survival fitness, infectivity, antigenicity, and, most worryingly, neutralization by ancestral vaccine- and infection-induced antibodies and sera [6]. The latter is carefully monitored as potential variants of concerns (VOCs) continue to emerge, such as the B1.1.7 (alpha) variant, first detected in the United Kingdom bearing an N501Y mutation in the receptor binding domain (RBD) of the spike (S) protein. This variant rapidly spread from September 2020 and in 2021, to all regions of the world, showing a reduced sensitivity to immune sera from Pfizer/BioNTech mRNA vaccinees and convalescents by factor 6 to 11, respectively [7,8]. Notably, the strain exhibited a higher reproduction rate than pre-existing variants resulting in higher viral loads and longer infectious periods [9,10]. In addition, the increased expression of inflammatory cytokines in nasal secretions was observed, indicating pronounced pathology [11].

At the same time, other VOCs have emerged in South Africa (B1.351, gamma variant) and in Brazil (P.1, beta variant), both of which contain mutations at positions K417N/T and N501Y in the RBD, the domain that is essential for binding to human ACE-2 (hACE-2) receptors [12,13]. Again, a significantly reduced neutralization efficiency of sera from convalescent plasma against SARS-CoV-2 variants bearing the above mentioned mutations was observed [13,14,15]. In 2021, the delta variant (B.1.617.2) emerged in Maharashtra, India, causing massive pressure to local healthcare and reached a prevalence of 87% by May 2021 [16]. It shows increased lung cell entry and higher viral shedding/transmissibility in both vaccinated and unvaccinated individuals [5,15]. As of July 2021, the delta lineage replaced most of the previous variants and produced a substantial number of vaccination breakthroughs [17,18].

Finally, in November 2021, a novel VOC of the pangolin lineage was characterized in South Africa, termed Omicron (B1.1.529). Omicron largely replaced the previously dominant delta variant and further evolved into sub-lineages with distinct geographical distribution (e.g., BA.1, BA.2, BA.2.12.1, and BA-3 to BA-5,) all of which display high transmissibility and immune escape from immunization sera [19,20]. Reinfections and breakthrough infections are more frequent than with previous variants [21] and the neutralization capacity of therapeutically used SARS-CoV-2 monoclonal antibodies clearly reduced [22]. Omicron variants display considerable antigenical distance to, e.g., the delta variant and represent a unique antigenic cluster showing increased infection but at a decreased risk of severe illness or hospitalization [23].

Originally, it was assumed that SARS-CoV-2 would have a minor propensity to mutate (an estimated 2 mutations per month), however, as the pandemic lingers on, new variants emerge continuously, which potentially could escape effective immunization. The relevance of single point mutations for the cellular innate immune response might be lower than for the generation of the humoral immune response, and epidemiological studies still reckon effective reduction of severe COVID-19 in fully vaccinated individuals [24]. At the same time, data draw a clearly less optimistic picture when it comes to mild to moderate diseases or delta variant transmission [25].

Thus, the search for robust preventive measures with low sensitivity towards genetic variations continues and might open up alternative strategies supplementing global vaccination endeavors. In particular, the observed vaccine’s inefficiency in preventing asymptomatic and milder COVID-19 illnesses and the virus transmission of newer variants call for additional solutions to reduce viral spreading since these cases account for the great majority of infections overall.

Antiviral and/or virucidal activity has been identified in medicinal plants but the isolation of compounds/substances that exert the respective activity, and manufacturing these into a product, remains challenging [26]. Plants produce a variety of substances as secondary plant products, i.e., for their own defense against pathogens including viruses. Since many derivatives of a single chemical structure are produced naturally, the problem of point mutations in the virus genome and possible viral evasion is expected to be reduced [27]. Likewise, virucidal activity against a wide series of common cold and highly pathogenic coronaviruses (CoV-229E, -MERS, SARS-1, and SARS-CoV-2) was previously demonstrated for an extract of *Echinacea purpurea* (Echinaforce^®^ extract, EF) in vitro [28]. Importantly, the clinical effects on enveloped and endemic coronaviruses corroborated preclinical findings leading to recommendations on the preventive use of Echinacea although modes of action (MOAs) are still poorly understood [29].

The current study aimed to investigate the antiviral/virucidal potential of the *Echinacea purpurea* extract regarding the propagation of actual SARS-CoV-2 VOCs and to explore possible MOAs. Virucidal activity against VOCs was studied at three different laboratories in parallel, using a standardized experimental protocol. We found strong and broad virucidal activity of EF extract against all investigated VOCs, including the predominant lineages of the alpha, beta, gamma, delta, and omicron variants as well as the isolates eta and the Scottish one. This study found that (i) the entry of pseudotyped viral particles bearing SARS-CoV-2 S protein was inhibited at similar EF concentrations as the wild type, along with (ii) data from molecular modelling clearly point towards an interaction of EF extract with the viral SARS-CoV-2 S protein. Cellular pre-treatment with EF further reduced the infectivity of SARS-CoV-2 virus, and immunocytochemical staining and ELISA point towards the possible inhibition of cellular proteases (e.g., TMPRSS-2), crucial for viral cell entry. Overall, our results together with data from Signer (2020) indicate a broad activity against coronaviruses and VOCs, highlighting multiple points of interference with viral infectivity by the EF extract and the contained known marker substances.

## 2. Materials and Methods

### 2.1. Test Material

In our experiments, we employed Echinaforce^®^ (A.Vogel AG, Roggwil, Switzer-land), a hydroethanolic extraction (65% *v*/*v* ethanol) of freshly harvested *Echinacea purpurea* produced according to Good Manufacturing Practices (GMP). The *Echinacea* herb and roots are extracted at a drug to extract ratio, DER 1:11 and 1:12, respectively, and combined at a final ratio of 95:5 (Batch no. 1053057). Ethanol was used for control matching the highest EF concentration in the individual experiment.

#### Cells and Viruses

African green monkey kidney epithelial cells clone C1008 (Vero-E6), ATCC (Cat# CRL-1586), HCT-8, ATCC (Cat# CCL-244) and HEK293T cells, ATCC (Cat# CRL-3216) were maintained with growth media (DMEM supplemented with 10% fetal calf serum (FCS, Invitrogen), 100 U/mL penicillin, and 0.1 mg/mL streptomycin (P/S, Thermo)). Human nasal epithelial cells (HNEpC, Promocell, Germany) and human bronchial epithelial cells (HBEpC, MatTek, USA) were cultivated as per the recommended protocol.

The following SARS-CoV-2 variant strains were used in this study: United Kingdom—B.1.1.7, South Africa—B.1.351, Brazil—P.1, India—B.1.617.2, Nigeria—B1.525, Scotland—AV.1, and Omicron—B.1.1.529, as well as the original SARS-CoV-2 (WUHAN-Hu-1) and the endemic human Coronavirus OC43. Stock viruses for SARS-CoV-2 and VOCs were propagated in Vero E6 cells and prepared as cell-free supernatants with titers ranging from 5 × 10^5^ to 10^6^ PFU (plaque-forming units) per ml and stored at −80 °C. Strains were titrated by standard plaque assay and OC43 stock was prepared in HCT-8 cells and titrated by CPE assay.

### 2.2. SARS-CoV-2 Pseudoparticle Generation

SARS-CoV-2 pseudoparticles were generated as previously described [30]. Briefly, lentiviral pseudotyped viruses are generated in HEK293T cells, which are transfected with a mixture of 0.6 µg p8.91 (HIV-1 gag-pol), 0.6 µg pCSFLW (lentivirus back-bone expressing a firefly luciferase reporter gene), and 0.5 µg pcDNA3.1 SARS-CoV-2 Spike D614 in OptiMEM with 10 µL polyethyleneimine 1 µg/mL (Sigma) as previously described [31]. The viral supernatant was harvested at 48 and 72 h post transfection, centrifuged to remove cell debris, and stored at −80 °C. The viral pseudo-particles were concentrated by overlaying the clarified supernatant on 20% sucrose and centrifugation at 23,000 rpm for 2 h at 4 °C. The purified pseudoparticles were then aliquoted and stored at −80 °C.

### 2.3. Cell Viability Assays

HEK293T cells were seeded at a density of 2 × 10^4^ cells per well in 100 µL DMEM supplemented with 10% FCS in a 96-well plate. Following 48 h of incubation at 37 °C, 2-fold serial dilutions of EF at a starting concentration of 200 µg/mL were added to the cell monolayer. Cells were examined microscopically for any sign of compound-induced cytotoxicity after 48 h. The cell viability was also determined via incubation with CellTiter-Glo^®^ (Promega) to measure the metabolic activity of inoculated and uninoculated cells for 1 h, and the luminescence was recorded after 10 min using GloMax^®^ Discover Microplate Reader (Promega).

Cytotoxicity of EF was determined by MTS assay [3-(4,5-dimethylthiazol-2-yl)-5-(3-carboxymethoxyphenyl)-2-(4-sulfophenyl)-2H-tetrazolium] (Promega, Madison, WI). Briefly, HCT-8, Vero E6, HBEpC, and HNEpC were cultured in 96-well plates and allowed to grow for 24 h at 37 °C in a 5% CO_2_ incubator. After 24 h, 2-fold serial dilutions of EF were added (100 to 0.39 µg/mL) to the cell monolayer, and incubated for 48 h at 37 °C and 5% CO_2_. Following incubation, 20 uL of MTS solution was added to each well and incubated for 1 h. The optical density was measured at 490 nm using a spectrophotometer. The percentage viability was calculated using the following formula: Viable cell number (%) = OD_490_ EF treated cells/OD_490_ (vehicle treated cells) × 100. Each assay was carried out in quadruplicate and the results were expressed as the mean ± standard deviation.

### 2.4. Virucidal Activity

Virucidal tests were carried out by the following facilities, which investigated the respective VOCs: B.1.1.7 (alpha), B1.351 (beta), and P.1 (gamma) (Pleschka, University Giessen, Germany); B.1.1.7 (alpha), B1.617.2 (delta), AV.1 (Scotland), B1.525 (eta), and B1.1.529 (omicron BA1) (Pariani, University Milano, Italy); and OC43 (Vimalanathan, University British Columbia, Canada). SARS-CoV-2 pseudotype assays were performed by the Viral Glycoproteins group, Pirbright Institute, United Kingdom. All facilities used the following experimental approach to ensure comparability of results with gradual modifications. Alternatively, plaque reduction and cytopathic endpoint dilutions were used as indicated.

SARS-CoV-2 mutant suspensions were prepared at 250 plaque forming units (pfu) per 80 µL of incomplete DMEM medium (DMEM without FBS). A total of 80 µL of test substance was added at concentrations of 50, 25, 10, 5, 2, and 0.1 µg/mL (dry mass) or 0.2% EtOH as control (ethanol concentration of the highest EF concentration), incubated for 1 h at room temperature (RT) and after 1 h incubation, samples were titrated and MIC_100_ and VC_50_ values of EF were determined by plaque reduction assay and CPE end point assay methods as follows.

#### 2.4.1. Plaque Reduction Assay

In 6-well tissue culture plates, Vero-E6 cells were subcultured with DMEM growth media and incubated overnight at 37 °C with 5% CO_2_. On the next day, 250 PFU/well in DMEM of the different SARS-CoV-2 strains were incubated with different nontoxic concentrations of the EF plant extract for 60 min at RT. Afterwards, the Vero-E6 cells monolayer was washed once with 1 × PBS and incubated with the different inoculum (virus/plant extracts mix) for 1 h at 37 °C, 5% CO_2_. Then, the inoculum was removed and cells were incubated with Avicel media (MEM supplemented with P/S, 2% FCS, 1.25% Avicel (RC-591, Dupont)) for 48 or 72 h at 37 °C, 5% CO_2_. The Avicel media was subsequently removed and cells were washed 3 × with PBS, fixed with 10% formalin for 60 min and stained (0.5% crystal violet, 20% methanol in dH2O) for 15 min at RT. Plaque titer (Plaque Forming Unit/mL) was calculated as mean number of plaques × dilution factor. This formula was used to calculate the titer of the virus suspension before the test with Echinacea and to verify the virucidal activity of Echinacea. To determine the virucidal concentration 50% (VC_50_), the plaque titers were calculated in percentage (% titer = (100/titer of untreated sample) × titer of extract/SC-treated sample) with the control (untreated) set as 100% using GraphPad Prism version 9 (GraphPad Software, San Diego, USA). The VC_50_ is defined as the compound concentration that inhibited virus-induced plaque by 50%.

#### 2.4.2. Cytopathic Effect

For the cytopathic effect (CPE) end-point method, Vero-E6 cells were grown in 96 well plates as described above to reached 95 to 100% confluency. Virus suspension was prepared at 100 PFU/100 µL in incomplete media and added to cells at the indicated dilutions prior to incubation at RT for 1 h. Growth media was removed and samples were transferred from 96-well plates to cell culture plates, returned to CO_2_ incubator then incubated for ± 72 h until CPE was examined visually. The MIC_100_ was the minimum concentration at which CPE was completely inhibited by the extract.

### 2.5. Pretreatment of Human Airway Epithelial Cells

Primary human nasal epithelial cells (HNEpC) and primary human bronchial epithelial cells (HBEpC) were purchased from PromoCell GmbH, Germany, and MatTek Inc, USA, respectively. HNEpC and HBEpC cells were seeded (1 × 10^4^/well) in 12-well plates and cultured in growth media free of antibiotics until 70% confluency was attained. Cells were pretreated with increasing concentrations of EF (5 to 80 µg/mL), and untreated wells received media alone or media with 0.2% ethanol for 24 h. After 24 h, supernatant was removed, cells were rinsed with PBS and infected with OC43 or SARS-CoV-2 at the MOI of 1 and incubated for 72 h at 34 °C and 37 °C, respectively, in 5% CO_2_; CPE was monitored microscopically. For infectious viral titer analysis, at 72 h post infection (hpi), cells were scraped from each well and collected with supernatant, and viral titer was determined on HCT-8 or Vero-E6 cells. Briefly, one day prior to infection, HCT-8 or Vero-E6 cells were seeded in 96-well plates at 1 × 10^4^ per well. Samples were serially diluted 2-fold with specific culture media containing 2% FBS and transferred to HCT-8 or Vero-E6 cells, in 6 parallel wells (100 µL/well) and incubated for 72 h. CPE was observed, recorded, and TCID 50/mL was calculated for each sample. The cytotoxicity of EF was tested in parallel on uninfected cells using the MTS assay (Promega).

### 2.6. Immunofluorescence Staining

HNEpC cells were cultured on coverslips and pretreated as indicated in Section 2.7. Pretreated HNEpC cells were gently washed with cold PBS 3 times and fixed with 10% formalin for 10 min at RT. Fixative was then removed and cells were washed with PBS 3 times and permeabilized with 0.1% saponin in PBS for 5 min; permeabilization buffer was removed and cells were washed 3 times with PBST for 5 min each. Nonspecific antigens were blocked with 1% BSA in PBST for 1 h at RT. Then, cells were incubated at 4 °C overnight with primary antibody (TMPRSS-2, Novus Biologicals cat# NBP1–20984) diluted to 5 µg/mL in blocking buffer. The next day, primary antibody was aspirated and cells were washed 3 times with PBST, 5 min each, then incubated with secondary antibody (Abcam, Ab150141) diluted to 1:250 in blocking buffer for 1 h at RT. Then, cells were washed with PBST and mounted with 10 µL of aqueous mounting medium, Fluoroshield DAPI (Abcam, Ab 104139) for nuclear staining. Images were captured on Ziess Axio observer Z1 inverted fluorescent microscope (Zeiss, Germany) The fluorescent intensity was calculated using Imagej software (open source).

### 2.7. Molecular Modeling and Molecular Docking Studies

The proper genome sequence of pandemic SARS-CoV-2 virus was retrieved from the National Centre for Biotechnology Information (NCBI) nucleotide database with the reference NC_045512.2. The available 3D crystal structures of various target SARS-CoV-2 proteins known as Papain Like protease (PLpro), 3C-like protease (3CLpro), RNA dependent RNA polymerase (RdRp), Spike (S) protein, human Angiotensin converting enzyme 2 (hACE2), and Janus kinase 2 (JAK2) were obtained from protein data bank (PDB) [32] with the accession code 6W9C, 5R7Z, 7BV2, 6M0J_E, 6M0J_A, and 2XA4, respectively. The optimum structures of other nonstructural proteins, namely NSP9, NSP13, NSP14, NSP15, and NSP16, were built using homology modeling with appropriate templates using the Swiss model [33] and I-TASSER web servers [34]. The most active regions of the proteins were figured out using the COACH meta-server [35] and outcomes were validated with the CASTp web server [36]. The antiviral and anti-inflammatory activities of compounds extracted from *Echinacea purpurea* against SARS-CoV-2 virus were examined by Molecular Docking studies using Schrodinger suite (Maestro) [37]. All the target proteins were processed by protein preparation wizard software (Schrodinger San Diago, CA, USA) and the ligands were constructed using LigPrep (Schrodinger San Diago, CA, USA). The protein structure was preprocessed in which the missing C- and N-terminal residues are capped and the missing side chain is filled. The overlapping hydrogens added to the proteins were optimized. The water molecules attached with the proteins are removed and the clean structure of the target proteins was minimized before docking. The tautomers of the ligands were generated, the specified chiralities of the ligands were retained, and at most 32 poses per ligand were generated using the LigPrep module. The OPLS3 force field with Extra Precision mode (XP) was performed to find the best position of ligands that adapt well in the cavity of the proteins.

### 2.8. Statistics

The experiments were conducted in triplicates and the given data are shown ± SEM or SD. One-way ANOVA followed by the Tukey multiple comparisons test was performed using GraphPad Prism version 9 (GraphPad Software, San Diego, CA, USA). The VC_50_ was determined using logarithmic interpolation.

### 2.9. Biosafety

All experiments performed with infectious VOCs were performed according to regulations for the propagation of BSL-3 viruses in a biosafety level 3 (BSL3) containment laboratory approved for such use by the respective local authorities.

## 3. Results

### 3.1. Cell Viability Tests

Cytotoxicity tests revealed that none of the tested EF concentrations showed a statistically significant difference in cell viability compared to cell controls. After 48 h of exposure to HNBEp and HBEpC cells, EF 100 µg showed 86% cell viability and EF 50 µg to 0.39 µg/mL displayed 100% cell viability compared to untreated cells. In Vero E6, HCT-8, and HEK 2 cells, EF 100 µg/mL to 0.39 µg/mL showed 100% viability.

### 3.2. Pretreatment of VOCs Prevents Viral Propagation at Low, Non-Toxic EF Concentrations

In a first approach to determine the virucidal potency of EF against VOCs, B1.1.7 (alpha), B.1.351.1 (beta), P.1 (gamma), B1.617.2 (delta), AV.1 (Scottish), B1.525 (eta), and B1.1.529 (omicron) variants were pretreated for 1 h with different, non-toxic concentrations (Singer et al., 2020) of EF (0 to 50 µg/mL) and subsequently, this inoculate was used to infect Vero E6 cells (250 PFU/well). Viral propagation was then assayed via plaque assay at 48 or 72 h post infection (pi). EF exhibited a very potent and broad virucidal effect against all investigated SARS-CoV-2 VOCs (Figure 1). The complete prevention of viral propagation was observed at EF concentrations equal and higher than 25 µg/mL for all investigated VOCs and across all laboratories (Figure 1). For all four laboratories, highly constant inhibitory concentrations were found with VC50 ranging from 5.37 to 12.03 µg/mL. One lab detected complete inhibition of P.1, B.1.351, and B.1.1.7 propagation at the lowest EF concentration of 1 µg/mL, which might be due to the fact that the virus stocks were derived from different sources for each lab. Results between labs were qualitatively highly consistent and show a strong inhibitory potential of EF extract irrespective of the different viral mutations.

#### 3.2.1. Pretreatment of Pseudoviruses Expressing Wild Type S Protein Prevents Viral Infection at Low, Non-Toxic Concentrations

SARS-CoV-2 pseudoparticles were generated via the lentiviral-based pseudotyping approach, as previously described [31]. EF showed a dose-dependent inhibitory activity against SARS-CoV-2 S protein-expressing pseudoparticles with an EC50 of 3.62 +/− 2.09 µg/mL (Figure 2). The fact that EF impaired the infectivity of both SARS-CoV-2 and SARS-CoV-2 pseudoparticles at comparable concentrations highlights the S protein as a potential target for EF activity. Interestingly, the infectivity of SARS-CoV pseudoparticles was also inhibited by EF but at an EC50 of 16.7 +/− 10.15 µg/mL, indicating a more potent activity towards SARS-CoV-2 over SARS-CoV.

#### 3.2.2. Pretreatment of Primary HNEpC and HBEpC Impairs Infection with OC-43 and SARS-CoV-2

Next to the preincubation of the viruses, primary HNEpC and HBEpC were preincubated with EF extract simulating a preventive treatment situation. EF at concentrations of 25 µg/mL completely reduced viral infections by ancestral SARS-CoV-2 (Hu-1) in both cell types as determined by CPE (Figure 3) end point assay after three days pi. Interestingly, in nasal epithelial cells infected with SARS-CoV-2, CPE was detected only 5 days pi; conversely, bronchial epithelial cells clearly showed CPE two days pi.

### 3.3. Molecular Docking (MD)

Herbal medicinal products typically contain a variety of pharmacologically active substances and in a first approach, several potentially active pharmacological substances were investigated for their interaction with well-known viral and cellular proteins in-volved in SARS-CoV-2 infection. Therefore, various alkylamide derivatives, caftaric acid, and 2-0-feruoyl-tartaric acid (totally 17 compounds) were tested for their docking potential against 12 different target proteins of SARS-CoV-2 virus (3CLpro, PLpro, RdRp, S-protein, NSP9, NSP13-16) and human cell proteins (ACE2, TMPRSS-2, and JAK2). The structure of the S protein and its receptor-binding domain (RBD, shown in red color) is represented in Figure 4. The RBD of the S protein contains, i.e., 18 amino acid residues (K417, G446, Y449, Y453, L455, F456, A475, F486, N487, Y489, Q493, S494, G496, Q498, T500, N501, G502, Y505). Looking at the surface of the S protein, the active region of the protein has a cave-like domain, allowing ligands to adapt and fit into this domain (as shown in Figure 4).

The binding affinities of all 17 compounds towards 12 different target proteins of SARS-CoV-2 virus are shown in Table 1. For the compound dodeca-2E,4E,8Z,10E-tetraensaure-isobutylamid belonging to the alkylamide group, a higher binding energy of −7.6 kcal/mol with the S protein was identified. This is depicted in Figure 5, which shows the hydrogen bonding interaction with the residues Gln 493 and Ser 494, Pi-Alkyl interaction with Tyr 453, Tyr 495, and Tyr 505, and van der Waals interaction with residues Arg 403, Tyr 449, Gly 496, and Asn 501.

Furthermore, MD analysis also demonstrated that alkylamides can interact with the RBD of the S protein. Caftaric acid and 2-0-feruloly-tartaric acid were both shown to have good binding affinity to the S protein and also to NSPs with 5.1/5.7 kcal/mol and 7.4/7.6 kcal/mol, respectively.

### 3.4. Echinacea’s Interaction with Pharmaceutical Targets for CoV Infection Process

Next, we investigated to what extent the molecular dynamic results obtained are applicable to a biological system and measured the effect of EF on the S protein inter-action with the cellular binding receptor ACE2. To this point, an S protein (RBD)-coated ELISA plate (SARS-CoV-2 NeutraLISA, EuroImmune, Germany), originally developed to measure neutralizing antibodies, was incubated with EF extract, which reduced the binding of biotinylated ACE2 receptors by 21.05 +/− 0.52% at concentrations of 50 µg/mL, *p* < 0.05 compared to the negative control (Figure 6). At least a partial blockade of the RBD could be deduced, albeit interactions with other S domains are expected to fully explain the inhibitory effects of EF on CoV infectivity.

Additionally, all compounds were tested as described in Section 3.4 against X-ray structures of the S protein containing the mutations from the alpha, beta, gamma, and delta VOCs and comparable binding energies as for wild type HU-1 were found. These data therefore indicate a very broad range of binding activity of the different compounds to the S protein of different SARS-CoV-2 strains/VOCs, which is largely permissive for point mutations. Relevant interactions with the S protein were further observed for other substances as well (e.g., caftaric acid), all of which appear to contribute to the overall bioactivity of EF.

### 3.5. Effect of EF on TMPRSS-2 Expression

Furthermore, EF affected another cellular component. As shown by immuno-histochemistry, treatment with EF extract at concentrations of 40 to 80 µg/mL resulted in the significantly reduced expression of TMPRSS-2 in primary nasal epithelial cells (Figure 7A), a result that was quantified by the Imagej software package (Figure 7B). No effects of EF on the protease furin expression (needed for S protein activation) could be detected, even at EF concentrations above 160 µg/mL (data not shown). Taken together, the results obtained from the MD analysis, the ACE2/S protein binding analysis, and the TMPRSS-2 expression study indicate a multifunctional MOA for EF, operating on various levels, i.e., partial interaction with the S protein, and concomitant inhibition of cellular receptors required for viral attachment and invasion of the host, as well as impaired expression of a cellular protease, which is essential for SARS-CoV-2 membrane fusion and virus infection.

## 4. Discussions

The COVID-19 pandemic has impressively demonstrated how viral pathogens can rapidly spill over from distant countries and spread world-wide within a few weeks time. Several vaccines using different immunization techniques were developed since and approximately one year after the genomic characterization of SARS-CoV-2. In January 2020, the first vaccines achieving regulatory approvals were available. Nevertheless, another year elapsed before broad vaccination coverage was accomplished, as in the case of the Israelian population attempting to achieve herd immunity. Many countries still have not reached immunization rates of more than 50%, almost 2.5 years since the onset of SARS-CoV-2. This delayed attainment of herd immunity gives the virus the possibility for further development through mutations and to evade containment by vaccines. In fact, viral variants and lineages develop faster than epidemiological studies are able to estimate real-life results for vaccination efficacy.

Thus, broadly effective and readily available antivirals are urgently needed, which are less susceptible to the spontaneous genetic variation frequently detected in viral respiratory pathogens not only in SARS-CoV-2. Multi-compound extracts derived from medicinal plants might provide an option here, as has been demonstrated for influenza viruses, known to quickly elicit resistance against oseltamivir treatment but not against treatment with *Echinacea purpurea* [27].

Our results together with earlier data from Signer (2020) infer a very broad virucidal activity of Echinaforce^®^ extract against coronaviruses that comprises the actually circulating SARS-CoV-2 VOCs as well. The inhibition of infectivity exerted upon the direct contact of virus with the extract, in addition, a preventive treatment of cells, provided a good level of protection against infection with the virus. Effective concentrations (VC50 < 12.03 µg/mL) varied marginally between different VOCs within the particular testing facilities. One laboratory generally measured virucidal effects at lower EF concentrations and a minimum concentration to inhibit 100% virus infectivity (MIC_100_) of 1 µg/mL for B.1.1.7, P.1, and B.1.351 virus variants possibly reflecting the fact that the virus stocks were derived from different sources for each lab. Our results are in good agreement with previous findings for other enveloped viruses, where similar MIC_100_ were found for various influenza A virus strains (H3N2, H5N1, H7N7, H7N9, or influenza B virus) at EF concentrations below 50 µg/mL [27,38].

Echinacea treatment of pseudoparticles specifically expressing SARS-CoV-2 spike D614 provided similar inhibitory effectiveness as with SARS-CoV-2 VOCs, suggesting an inhibitory action on the S protein, which may explain a MOA at the viral entry level as observed for influenza viruses [27]. This hypothesis was further substantiated by ELISA experiments and molecular docking calculations revealing relevant binding affinities for a series of compounds in *Echinacea purpurea*, including alkylamides, its derivatives, or caftaric acid. The fact that different substances show a certain level of binding not only to S protein but also to the serine protease (TMPRSS-2) or non-structural proteins (NSPs) indicates that the complexity of the whole extract might deliver the observed broad range inhibition in a concerted manner. Notably, different binding sites were identified for different compounds indicating synergistic effects of the complex substance mixture. As of yet, it appears impossible and meaningless to trace the antiviral activity down to a single substance in EF while still retaining its full spectrum benefits.

The question remains regarding the clinical relevance of the presented findings, which, together with data from [28], infer a generic activity against coronaviruses overall. Evidence was gathered from three clinical studies, two on endemic CoVs and one comparative trial investigating SARS-CoV-2. For 4 months, Jawad studied the effect of Echinaforce^®^ on the incidence of respiratory tract infections in 755 volunteers. In this randomized, double blind, placebo-controlled, clinical study, 54 viral infections occurred, of which 21 were caused by endemic coronaviruses (CoV-229E, CoV-HKU1, and HCoV-OC43) in the Echinacea group in contrast to 33 coronavirus infections in the placebo group [39]. Statistical significance was reached for the overall incidence of enveloped viruses (including coronaviruses, odds ratio OR = 0.49; *p* = 0.0114) rather than on the level of particular pathogens. Ogal and colleagues investigated the outcome of 4 months of preventive treatment with the same extract that we used and found a 98.5% decreased virus concentration in nasal secretions (*p* < 0.05) in comparison with control treatment [29]. Again, significantly fewer enveloped virus infections (including coronaviruses) occurred with EF treatment overall (OR = 0.43; *p* = 0.0038), substantiating the relevance of antiviral effects in vivo. Of note, in both studies participants kept the Echinacea formulation in the mouth for a few seconds prior to swallowing to enhance local antiviral effects. A very recent study was carried out during the COVID-19 pandemic, which routinely collected naso/oropharyngeal samples during 5 months of preventive treatment with EF extract or control. Here, Echinaforce^®^ significantly reduced SARS-CoV-2 incidences from 14 to 5 infections (RR = 0.42, *p* = 0.046) and overall viral loads by more than 2.1 logs, corresponding with a > 99% viral reduction (*p* = 0.04). Finally, the time to become virus-free was significantly shortened.

The above presented data clearly indicate the clinical relevance of in vitro antiviral effects observed for *Echinacea purpurea* extract. This is highly important because all coronaviruses tested so far were inactivated by EF extract and the applicability of results to the in vivo situation would imply a general protective benefit not only against endemic strains, but also to newly occurring coronaviruses including their VOCs.

## 5. Conclusions

EF extract demonstrated broad and stable antiviral activity across seven tested VOCs, which is likely due to the constant affinity of the contained phytochemical marker substances to all spike variants. A potential interaction of EF with TMPRSS-2 would partially explain the cell protective benefits of the extract by the inhibition of membrane fusion and viral entry. EF may therefore offer a supportive addition to vaccination endeavors in the control of existing and future virus mutations.

## Figures and Tables

**Figure 1 microorganisms-10-02145-f001:**
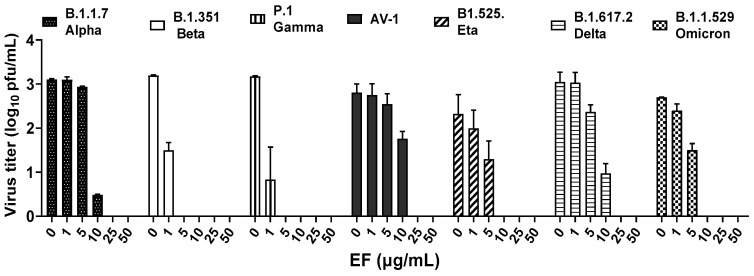
EF extract fully inhibited infectivity of all investigated variants of concerns (VOCs) at concentrations ≥ 25 µg/mL.

**Figure 2 microorganisms-10-02145-f002:**
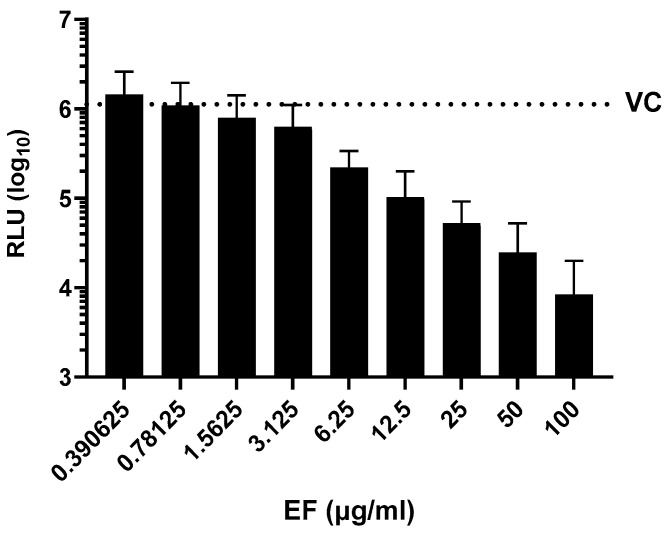
Inhibition of SARS-CoV-2 pseudoparticle infectivity by EF extract. Serial dilutions of EF extract were incubated with SARS-CoV-2 pseudoparticles for 1 h showing a dose-dependent inhibition of SARS-CoV-2 infectivity as measured by intracellular luciferase expression in mean relative light units (RLU). VC represents the RLU value of SARS-CoV-2 pseudoparticles-infected untreated cells. Data represent mean values ± standard deviations from three independent experiments performed in quadruplicates per experiment.

**Figure 3 microorganisms-10-02145-f003:**
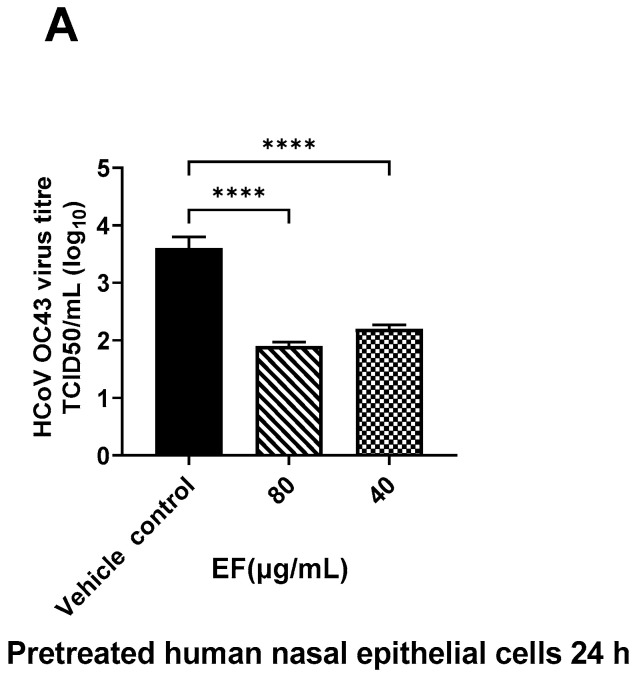
Infection preventive (antiviral) effect of EF on pretreated human airway epithelial cells. EF prevented OC43 (**A**) and SARS-CoV-2 (HU-1, (**B**,**C**)) infection dose-dependently in pretreated HNEpC and HBEpC. Cells were pretreated with EF for 24 h and infected with virus at MOI of 1. Viral titer was measured at 72 h pi. Data resulted from 3 independent experiments. One-way ANOVA with Bonferroni multiple comparison correction was performed (****: *p* ≤ 0.0001).

**Figure 4 microorganisms-10-02145-f004:**
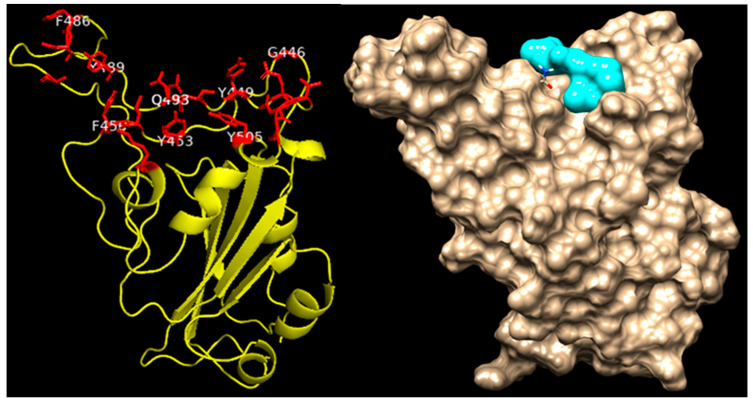
Structure of Spike protein of SARS-CoV-2 virus. The receptor binding domain (RBD) of the Spike protein is indicated in red color stick/blue.

**Figure 5 microorganisms-10-02145-f005:**
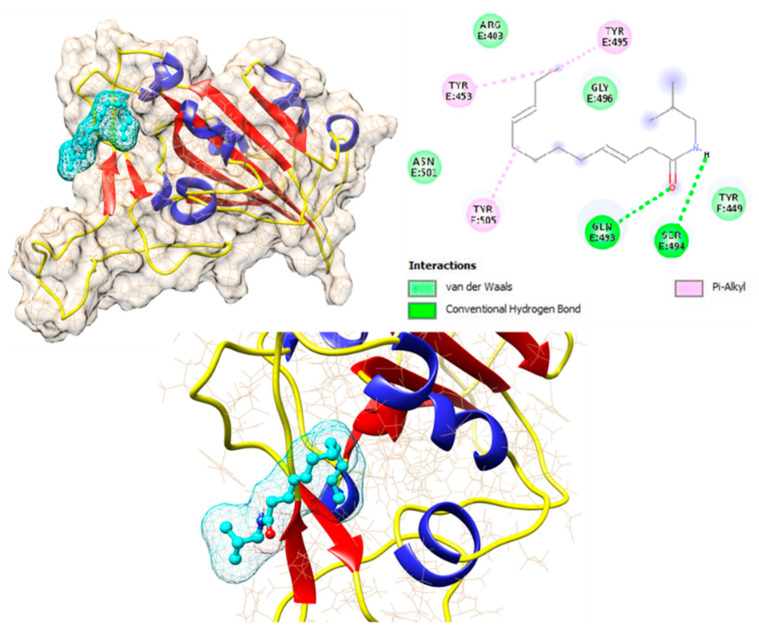
Interaction of dodeca-2E,4E,8Z,10E-tetraensaure-isobutylamid with wild type Spike protein of SARS-CoV-2 protein.

**Figure 6 microorganisms-10-02145-f006:**
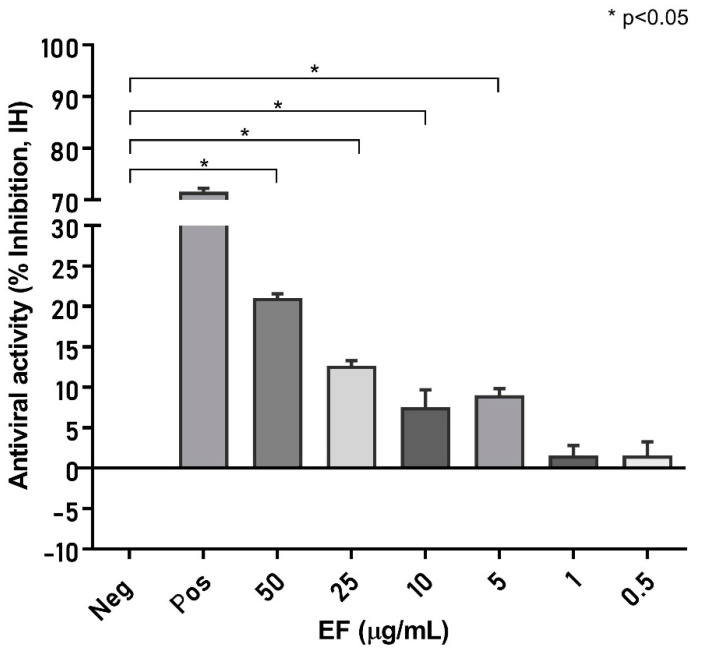
EF affects S protein binding to ACE2 receptor. S protein (RBD)-coated ELISA plates were incubated with increasing EF concentrations and subsequently the binding of biotinylated ACE2 receptors was quantified by measurement of chemiluminescence (% Inhibition, IH). As a positive control, ACE2 was co-incubated with immunized patient sera, which inhibited more than 71.5% of ACE2 receptor binding. EF at 50 µg/mL inhibited S protein binding, up to 21.0% to ACE2 (*p* < 0.05).

**Figure 7 microorganisms-10-02145-f007:**
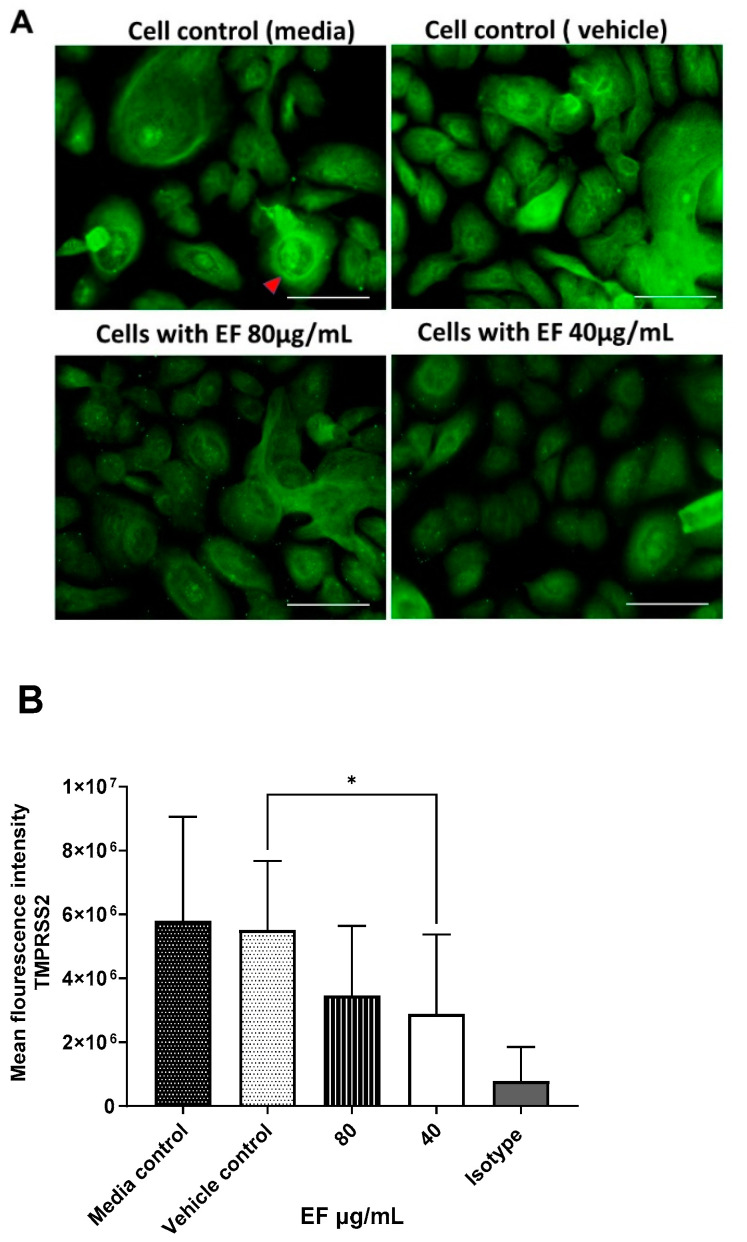
Inhibitory effect EF on TMPRSS-2 expression in EF pretreated HNEpC cells. Cultured HNEpC cells were treated with 80 and 40 µg/mL EF for 48 h. (**A**) Surface and intracellular immunofluorescent staining using an TMPRSS-2 antibody. (**B**) Quantification of the TMPRSS-2 signal. Graphs represent mean inflorescence values and ± SD of three independent experiments (* *p* < 0.05). Fluorescent intensity was calculated using Imagej software package (open source). Scale bar = 100µm.

**Table 1 microorganisms-10-02145-t001:** The binding affinity of compounds of *Echinacea purpurea* against various target proteins of SARS-CoV-2 virus and of human cells in kcal/mol using Schrödinger XP Glide software package.

Name of Echinacea Purpurea Compounds	3CLpro	PLpro	RdRp	S-Protein	NSP9	NSP13	NSP14	NSP15	NSP16	ACE2	TMPRSS-2	JAK-Janus Kianse
Undeca-2E,4Z-dien-8,10-diinsaure-isobutylamid	−5.7	−4.6	−5	−5.4	−4.6	−5.5	−6.5	−5.9	−5.6	−4.3	−5	−5.8
Undeca-2Z,4E-dien-8,10-diinsaure-isobutylamid	−5.7	−4.7	−5	−5.3	−4.4	−5.6	−7	−4.8	−5.8	−5.9	−5.2	−5.8
Dodeca-2E,4Z-dien-8,10-diinsaure-isobutylamid	−5.1	−5.6	−5.2	−5.2	−4.5	−5.8	−7.4	−5.8	−6	−5.6	−5	−6.1
Undeca-2E,4Z-dien-8,10-diinsaure-2-methybutylamid	−5.3	−5.3	−5.1	−5.2	−4.5	−5.7	−6.7	−5.6	−6.1	−5.9	−5.5	−5.8
Dodeca-2E,4E,10E-trien-8-insaure-isobutylamid	−5.2	−5.4	−5.4	−5.3	−4.9	−5.9	−7.3	−6.1	−5.8	−5.7	−5.6	−6.1
Trideca-2E,7Z-dien-10,12-diinasaure-isobutylamid	−4.8	−4.4	−5.2	−5.3	−4.5	−5.6	−6.6	−5.8	−5.6	−4.4	−5.3	−5.9
Dodeca-2E,4Z-dien-8,10-diinasaure-2-methybutylamid	−5.2	−5.4	−5.2	−6.9	−4.1	−6.4	−7.1	−5.5	−5.8	−6.3	−5	−6.3
Dodeca-2E,4E,8Z,10E-tetraensaure-isobutylamid	−5.2	−4.7	−5.2	−7.6	−4.7	−5.9	−7.6	−6.2	−6.1	−4.7	−5.2	−5.8
Dodeca-2E,4E,8Z,10Z-tetraensaure-isobutylamid	−5	−5.3	−5.1	−6.0	−4.6	−5.9	−6.6	−6.2	−5.8	−5.5	−5.4	−5.9
Dodeca-2E,4E,8Z-triensaure-isobutylamid	−4.9	−5.3	−5.3	−6.0	−4.3	−5.7	−6.3	−5.9	−5.6	−4.4	−5.3	−5.5
Dodeca-2E,4E-diensaure-isobutylamid	−4.7	−4.3	−4.8	−5.0	−4.3	−5.5	−6.5	−5.1	−5.6	−5.6	−5.3	−5.7
7-Hydroxy-Dodeca-2E,4E,8Z,10E-tetraensaure-isobutylamid	−5.4	−4.8	−5.7	−4.8	−4.5	−6	−7.1	−6.5	−6.2	−4.8	−5.8	−5.9
Undeca-2E,4Z-dien-8,10-diinsäure-isobutylamide	−5.2	−5	−5	−5.6	−4.8	−5.7	−6.9	−5.9	−5.7	−5.6	−5.3	−5.8
Pentadeca-2E,9Z-dien-12,14 diinsaure-isobutylamide and hydroxylated derivates	−5.1	−4.3	−4.6	−4.9	−4.3	−5.8	−6.8	−5.2	−5.4	−4.8	−4.9	−5.7
Trideca-2E,7Z-dien-10,12-diinsäure-isobutylamid	−4.9	−5.3	−4.8	−5.5	−4.6	−6.3	−7.3	−5.6	−5.5	−6	−5.5	−6
Caftaric acid	−6.8	−6.4	−6.6	−5.1	−5.1	−7.1	−7.4	−6.6	−7.3	−6.4	−6.6	−6.8
2-0-feruloly-tartaric acid	−6.4	−6.3	−6.6	−5.7	−4.6	−7.1	−7.6	−6.2	−7.4	−5.9	−6.4	−6.6

## Data Availability

Data presented in this study are principally contained in the manuscript and available upon reasonable request from the corresponding author.

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
