# Peer review of "Broad Antiviral Effects of Echinacea purpurea against SARS-CoV-2 Variants of Concern and Potential Mechanism of Action"

_microorganisms, 2022, doi:10.3390/microorganisms10112145_

Round 1

Reviewer 1 Report

Dear Editor,

The submitted manuscript presents the effects of Echinaforce® extract, a hydroethanolic extraction of freshly harvested Echinacea purpurea against wild type SARS-CoV-2 (Hu-1), SARS-CoV-2 variants of concern (VOCs) and seasonal coronavirus OC43. Echinacea extracts are best known as an over-the-counter herbal remedies for dealing with the common cold, flu and flu-like illness.

Till now, SARS-CoV-2 has infected more than 600 mln causing more than 6.5 mln deaths. Although more than a dozen effective vaccines and a few selective antivirals are already in use, there is a constant need for a broad-spectrum and costly affordable remedy capable of shortening the duration of illness and relieving the symptoms. Echinacea extracts have undergone several randomized blind placebo controlled trials studying their effect on seasonal coronaviruses, and SARS-CoV-2, as well, with promising results.

The presented manuscript makes an attempt to reveal the potential mechanism of action of the hydroethanolic extraction of Echinacea along with providing chemotherapeutic in vitro data on its virucidal and antiviral effect. Affinity of the numerous compounds present in Echinaforce® extract, to the viral spike protein and possible interaction with cellular protease participating in the process of virus entry into the cell, is suspected as a mechanism of action.

In this respect, the proposed manuscript provides important information.

Nevertheless, I have some major concerns and minor remarks, which are as follows:

Major Remarks:

1.    Title

I am not sure that the term “antiviral” is appropriately used in the title and further throughout the text. The experimental set-up is rather for testing virucidal activity, directed mainly against extracellular virions.

 2.    Abstract, line 41.

Usually, SARS-CoV-2, with the exception of its Omicron variant of concern, does not enter the cell by receptor-mediated endocytosis. Rather the virus uses the cell surface fusion pathway triggered by cellular TMPRRS2. Of course, the virus may use the endocytosis pathway mediated by cellular cathepsin L within the endosome, also, and in fact, Omicron does predominantly that (Willet, B. et al., 2022, Nature Microbiology, 7, 1161). As earlier variants of concern along with the ancestral Hu-1 virus are included in the research, I would suggest omission/replacement of the term “inhibition of endocytosis” in the abstract.

3.      Introduction

In general, the Introduction is a bit out-of-date. Some of the variants, referred to in the Present tense, have already vanished. For example, Delta is not predominant at the moment. Omicron took the lead. Please mind Minor Remarks 3, 5, and 6.

4.      Introduction, lines 102-104.

The cited authors, Signer et al. (Ref.22) reveal virucidal, not antiviral effects of Echinaforce® extract: “post-infection treatment had only a marginal effect on virus propagation at 50 μg/mL”

5.      Materials and Methods, 2.2. SARS-CoV-2 pseudoparticle generation

The generation of SARS-CoV pseudoparticles is not described, although results on pseudovirus particles of SARS-CoV are reported in section 3.1.2., lines 285-287.

 6.      Materials and Methods, 2.4. Virucidal activity.

How it was proceeded following the 1-hour contact time of equal volumes of virus suspension and Echinaforce® extract? Were these samples titrated and if yes, how were they titrated – by the plaque assay or by the end-point dilution CPE assay?

 7.      Materials and Methods, lines 191-196.

The description of the CPE assay should not be under the Plaque Reduction Assay subtitle. Please describe the procedure more thoroughly. When virus suspension was added (line 193), was the growth medium discarded before that? When and how was Echinaforce® extract added? Explain the need for and how the transfer from 96-well plates to cell culture plates has been done (lines 194-195). Explain what you mean by incubation for ±72 h. ±?

 8.      Results

Data from the cell viability assays carried out in this research are not presented. What are the maximal tolerated concentrations (MTC) and the 50% cytotoxic concentrations (CC50) of Echinaforce® extract for HEK293T cells? Are toxicity assays carried out for the epithelial nasal and bronchial cells?

 9.      Results, section 3.1.

In fact, these results do not present pure antiviral activity. They present rather virucidal activity because the inoculated virus is pretreated for an hour with Echinaforce® extract. But for scoring virucidal activity one should titrate treated and untreated extracellular virus. Here, there is no titration of samples. First, there is a virus suspension that is mixed with an equal volume of Echinaforce® extract at different concentrations. Then this mixture is inoculated on cell monolayers and scored by the plaque assay. The number of plaques formed by the treated and untreated extracellular virus is scored. Actually, this is not EC50, because extracellular virions are treated before the first stage of the virus replication cycle, i.e. the adsorption stage. And EC50 is a parameter scoring the effect on virus replication in cells, not on extracellular virions.

10.  Results, section 3.1.3.

Are experiments carried out with the variants of concern? Or they are carried out only with OC43 and the ancestral Hu-1?

On lines 299-300 it is written that 20 μg/mL completely reduces viral infection by Hu-1 on day 3. But in Fig. 3 results from day 3 are not presented. Only results from day 1 are presented. What is more, on the figures the bars are not for a concentration of 20 μg/mL, but for 25 μg/mL.

Results for days 2 and 3 h p.i. are not presented. These time points (24, 48 and 72 h p.i.) are described in Materials and Methods, section 2.5. 

 Minor Remarks:

1.    Introduction, lines 47-48.

Error-prone viral RNA polymerase solely did not lead to the occurrence of variants. Rather the vast circulation among hosts aided the selection of mutants. What is more, coronavirus polymerase is the most fidelious one thanks to the exonuclease activity of NSP14.

2.    Introduction, line 56.

Newer variants of concern are capable of evading not only vaccine-induced antibodies but infection-induced ones, as well.

 3.    Introduction, line 75.

Delta is no more predominant in the USA.

4.    Introduction, line 78.

I think Omicron was first spotted in South Africa, not in the USA (Viana, R et al., 2022. Nature 603:679–686).

 5.    Introduction, line 80.

Omicron sublineages are far more numerous today.

 6.    Introduction, line 89.

The Omicron variant of concern should be mentioned here, also.

 7.    Introduction, line 119.

I would suggest inserting “cellular” proteases.

 8.    Introduction, line 120.

I would suggest restraining the use of the term “antiviral” since the observed effect is not, in fact, selective inhibition of virus replication.

9.    Materials and Methods, line 176

It should be subcultured.

10.    Materials and Methods, line 186

Define plaque titer. Usually, virus suspensions are titrated.

11.    Materials and Methods, line 204

I suppose cells were rinsed with PBS.

 12.    Materials and Methods, line 208.

HCT cells are not described in Materials and Methods.

 13.    Materials and Methods, lines 212-213

Cytotoxicity assays are described earlier in section 2.3. Is the MTS assay mentioned there?

 14.    Materials and Methods, line 215

In fact, Section 2.7 (end of the line) deals with molecular docking and does not describe pre-treatment.

 15.    Materials and Methods, line 216

Use the already introduced abbreviation for human nasal epithelial cells.

 16.    Results, section 3.1.3.

Use the already introduced abbreviation for human nasal and bronchial epithelial cells.

 17.    Discussion, line 412.

Describe MIC100.

 In conclusion, I would suggest a major revision of the manuscript before accepting it for publication.

Author Response

Dear Reviewer,

We would like to thank you for your valueable recommandations and comments. We have revised our manuscript according to your suggestions and please find our responses to your comments in the below attached file.

Thank you.

Sincerely,

Selvarani Vimalanathan  , Mahmoud Shehata , Kannan Sadasivam , Serena Delbue , Maria Dolci , Elena Pariani , Sarah D’Alessandro , Stephan Pleschka.

Reviewer 2 Report

I congratulate the authors for the research that is of great relevance to the scientific community and to the world population.

Below, I quote small changes to improve the manuscript.

1 - The article summary is too long. I suggest rewriting and making it more objective.

2 - The keywords are repetitions of the title. I suggest replacing it with other words that can draw attention to the reading of the text.

3 - Lines 110 to 111: Virucidal activity against VOCs was studied at 3 different laboratories in parallel, using a standardized experimental protocol. First question, replace 3 with three. Second question, why were the tests performed in three different labs?

Author Response

Dear Reviewer,

Thank you for your suggestions and comments, and we have revised our manuscript accordingly. We have attached a file below with our responses.

Thank you.

Sincerely,

Selvarani Vimalanathan  , Mahmoud Shehata , Kannan Sadasivam , Serena Delbue , Maria Dolci , Elena Pariani , Sarah D’Alessandro , Stephan Pleschka 

Reviewer 3 Report

This manuscript by Selvarani et al. "Broad antiviral effects of Echinacea purpurea against SARS-CoV-2 variants of concern and potential mechanism of action" is a good reasearch according with COVID-19 pandemic and their variants.. The paper discusses key features of these plant and molecules using  some viruses.  Overall, I think the information provided in this paper is good and interesting nowdays.  I do recommend this manuscript for publication.

Author Response

Dear Reviewer,

Thank you so much for your valuable comments and recommendation.

We are submitting the revised manuscript for acceptance.

Sincerely,

Selvarani Vimalanathan , Mahmoud Shehata , Kannan Sadasivam , Serena Delbue , Maria Dolci , Elena Pariani , Sarah D’Alessandro , Stephan Pleschka 

Round 2

Reviewer 1 Report

Dear Editor,

The authors have addressed diligently all my comments and remarks and did all the due changes in the text. I have no further remarks or considerations.